# Training Data Generating Networks: Shape Reconstruction via Bi-level Optimization

**Biao Zhang & Peter Wonka**
KAUST
{biao.zhang, peter.wonka}@kaust.edu.sa

## Abstract

We propose a novel 3d shape representation for 3d shape reconstruction from a single image. Rather than predicting a shape directly, we train a network to generate a training set which will be fed into another learning algorithm to define the shape. The nested optimization problem can be modeled by bi-level optimization. Specifically, the algorithms for bi-level optimization are also being used in meta learning approaches for few-shot learning. Our framework establishes a link between 3D shape analysis and few-shot learning. We combine training data generating networks with bi-level optimization algorithms to obtain a complete framework for which all components can be jointly trained. We improve upon recent work on standard benchmarks for 3d shape reconstruction.

## 1 Introduction

Neural networks have shown promising results for shape reconstruction (Wang et al., 2018; Groueix et al., 2018; Mescheder et al., 2019; Genova et al., 2019). Different from the image domain, there is no universally agreed upon way to represent 3d shapes. There exist many explicit and implicit representations. Explicit representations include point clouds (Qi et al., 2017a;b; Lin et al., 2017), grids (Wu et al., 2015; Choy et al., 2016; Riegler et al., 2017; Wang et al., 2017; Tatarchenko et al., 2017), and meshes (Wang et al., 2018; Hanocka et al., 2019). Implicit representations (Mescheder et al., 2019; Michalkiewicz et al., 2019; Park et al., 2019) define shapes as iso-surfaces of functions. Both types of representations are important as they have different advantages.

In this work, we set out to investigate how meta-learning can be used to learn shape representations. To employ meta-learning, we need to split the learning framework into two (or more) coupled learning problems. There are multiple design choices for exploring this idea. One related solution is to use a hypernetwork as the first learning algorithm that produces the weights for a second network. This approach (Littwin & Wolf, 2019), is very inspiring, but the results can still be improved by our work. The value of our work are not mainly the state-of-the-art results, but the new framework that enables the flow of new ideas, techniques, and methods developed in the context of few-shot learning and meta-optimization to be applied to shape reconstruction.

In order to derive our proposed solution, we draw inspiration from few-shot learning. The formulation of few-shot learning we use here is similar to Lee et al. (2019); Bertinetto et al. (2019), but also other formulations of few shot learning exist, *e.g.*, MAML (Finn et al., 2017). In few-shot learning for image classification, the learning problem is split into multiple tasks (See Fig. 1). For each task, the learning framework has to learn a decision boundary to distinguish between a smaller set of classes. In order to do so, a first network is used to embed images into a high-dimensional feature space. A second learner (often called *base learner* in the literature of meta learning approaches for few-shot learning) is then used to learn decision boundaries in this high-dimensional feature space. A key point is that each task has separate decision boundaries. In order to build a bridge from few-shot learning to shape representations, we identify each shape as a separate task. For each task, there are two classes to distinguish: inside and outside. The link might seem a bit unintuitive at first, because in the image classification case, each task is defined by multiple images already divided into multiple classes. However, in our case, a task is only associated with a single image. We therefore introduce an additional component, a training data generating network, to build the bridge to few-shot learning (See Fig. 2). This training data generating network takes a single image as input and

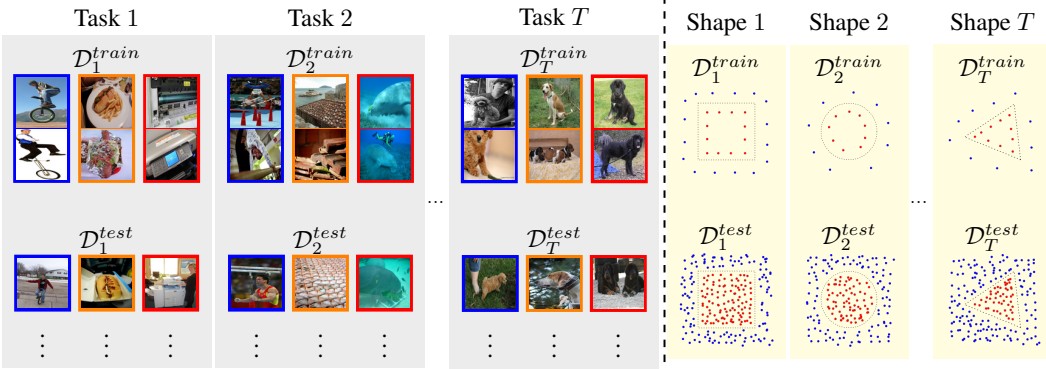

Figure 1: **Few-shot learning v.s. shape representation. Left (gray):** few-shot classification (images taken from miniImageNet). We show 2-shot-3-way few-shot classification in which every training task contains 3 categories (shown in blue, orange and red bounding boxes) and 2 training samples in each category. Thus we use $2 \times 3 = 6$ training samples in $\mathcal{D}_t^{train}$ to build classifiers which are expected to work well on $\mathcal{D}_t^{test}$. **Right (yellow):** shape representation. Shape surfaces are shown as dotted lines. Our proposed training data generating networks can encode a shape $t$ as labeled set of points with labels blue or red to provide a training set $\mathcal{D}_t^{train}$. The learning framework uses the training sets to build the surfaces (classification boundaries) which can be evaluated by densely sampling the space, *i.e.*, $\mathcal{D}_t^{test}$.

outputs multiple 3D points with an inside and outside label. These points define the training dataset for one task (shape). We adopt this data set format to describe a task, because points are a natural shape representation. Similar to the image classification setting, we also employ an embedding network that maps the original training dataset (a set of 3D points) to another space where it is easier to find a decision boundary (a distorted 3D space). In contrast to the image classification setting, the input and output spaces of the embedding network have a lot fewer dimensions, i.e. only three dimensions. Next, we have to tackle the base learner that takes a set of points in embedding space as input. Here, we can draw from multiple options. In particular, we experimented with kernel SVMs and kernel ridge regression. As both give similar results (In meta learning (Lee et al., 2019) we can also find the same conclusion), we opted for ridge regression in our solution due to the much faster processing time. The goal of ridge regression is to learn a decision boundary (shape surface) in 3D space describing the shape.

Our intermediate shape representation is a set of points, but these points cannot be chosen arbitrarily. The set of points are specifically generated so that the second learning algorithm can process them to find a decision boundary. The second learning algorithm functions similar to other implicit shape representations discussed above. The result of the second learning algorithm (SVM or ridge regression) can be queried as an implicit function. Still the second learning algorithm cannot be replaced by other existing networks employed in previous work. Therefore, among existing works, only our representation can directly use few-shot learning. It does not make sense to train our representation without a few-shot learning framework and no other representation can be directly trained with a few-shot learning framework. Therefore, all components in our framework are intrinsically linked and were designed together.

**Contributions.**

- We propose a new type of shape representation, where we train a network to output a training set (a set of labeled points) for another machine learning algorithm (Kernel Ridge Regression or Kernel-SVM).

- We found an elegant way to map the problem of shape reconstruction from a single image to the problem of few-shot classification by introducing a network to generate training data. We combined training data generating networks with two meta learning approaches (R2D2 (Bertinetto et al., 2019) and MetaOptNet (Lee et al., 2019)) in our framework. Our work enables the application of ideas and techniques developed in the few-shot learning literature to the problem of shape reconstruction.

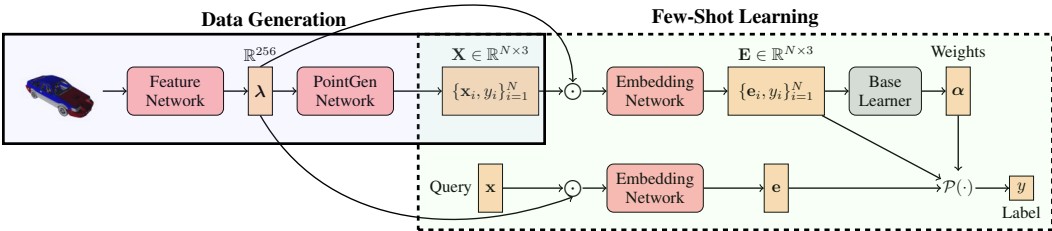

Figure 2: **Pipeline.** Networks with trainable parameters are shown in red boxes with round corners. Outputs are shown in yellow boxes. The base learner is a machine learning algorithm shown in the gray box. Arrows show how the data flows in the network. The sign $\odot$ means we are concatenating multiple inputs.

- We validate our model using the problem of 3d shape reconstruction from a single image and improve upon the state of the art.

## 2 RELATED WORK

**Neural implicit shape representations.** We can identify two major categories of shape representations: *explicit* representations, where a shape can be explicitly defined; and *implicit* representations, where a shape can be defined as iso-surface of a function (signed distance function or indicator function). In the past decade, we have seen great success with neural network based explicit shape representations: voxel representations (Wu et al., 2015; Choy et al., 2016; Riegler et al., 2017; Wang et al., 2017; Tatarchenko et al., 2017), point representations (Qi et al., 2017a;b; Fan et al., 2017; Lin et al., 2017), and mesh representations (Wang et al., 2018; Hanocka et al., 2019)). On the other hand, modeling implicit representations with neural networks has been a current trend, where usually a signed distance function or indicator function is parameterized by a neural network (Mescheder et al., 2019; Chen & Zhang, 2019; Michalkiewicz et al., 2019; Park et al., 2019). More recent works learn a network that outputs intermediate parameters, *e.g.* CvxNet (Deng et al., 2019) and BSP-Net (Chen et al., 2019) learns to output half-spaces. We propose a novel type of shape representation, where the model outputs a training set of labeled points.

**Few-shot learning.** There are two common meta learning approaches for few-shot learning: metric-based (Koch et al., 2015; Vinyals et al., 2016; Snell et al., 2017), which aims to learn a metric for each task; optimization-based (Ravi & Larochelle, 2017; Finn et al., 2017; Nichol et al., 2018), which is designed to learn with a few training samples by adjusting optimization algorithms. These approaches commonly have two parts, an embedding model for mapping an input to an embedding space, and a base learner for prediction. Qiao et al. (2018); Gidaris & Komodakis (2018) train a hypernetwork (Ha et al., 2016) to output weights of another network (base learner). R2D2 (Bertinetto et al., 2019) showed that using a light-weight and differentiable base learner (*e.g.* ridge regression) leads to better results. To further develop the idea, MetaOptNet (Lee et al., 2019) used multi-class support vector machines (Crammer & Singer, 2001) as base learner and incorporated differentiable optimization (Amos & Kolter, 2017; Gould et al., 2016) into the framework. Lee et al. (2019) also shows it can outperform hypernetwork-based methods. In our work, we propose a shape representation that is compatible with a few-shot classification framework so that we can utilize existing meta learning approaches. Specifically, we will use ridge regression and SVM as the base learner. The most relevant method to ours is Littwin & Wolf (2019) which adapts hypernetworks. However, as we discussed above, differentiable optimization methods (Bertinetto et al., 2019; Lee et al., 2019) are generally better than hypernetworks. Besides that, meta learning has been applied to other fields in shape analysis, *e.g.*, both Sitzmann et al. (2020) and Tancik et al. (2021) propose to use MAML-like algorithms (Finn et al., 2017) to learn a weight initialization.

## 3 METHOD

The framework is shown in Fig. 2. The network is mainly composed of 3 sub-networks. The Feature Network maps an input image to feature space. The resulting feature vector $\boldsymbol{\lambda}$ is then decoded by the Point Generation Network to a labeled point set $\{\mathbf{x}_i, y_i\}_{i=1}^N$. The point set will be used as training data in our base learner later. After that the Embedding Network projects the point set into embedding space. The projected points $\mathbf{e}_i$ and the labels are taken as the input of a binary classifer (ridge regression or SVM) parameterized by $\boldsymbol{\alpha}$. Finally, the framework is able to output the inside/outside label $y$ of a query point $\mathbf{x}$ by projecting it into the embedding space and feeding it to the binary classifier.

In the following subsections, we describe our method in more detail. First, we introduce the background of meta learning approaches for few-show learning (Sec. 3.1) and establish a link between single image 3D reconstruction and few-shot learning (Sec. 3.2). We propose a problem formulation inspired by few-shot learning (Sec. 3.3) and propose a solution in the following subsections. Specifically, we apply recently developed differentiable optimization.

### 3.1 BACKGROUND

**Supervised learning.** Given training set $\mathcal{D}^{train} = \{\mathbf{x}_i, y_i\}_{i=1}^N$, supervised learning learns a predictor $y = \mathcal{P}(\mathbf{x})$ which is able to predict the labels of test set $\mathcal{D}^{test} = \{\mathbf{x}_i, y_i\}_{i=1}^M$ (assuming both $\mathcal{D}^{train}$ and $\mathcal{D}^{test}$ are sampled from the same distribution).

**Few-shot learning.** In few-shot learning, the size $N$ of the training set is typically small. The common learning algorithms on a single task usually cause problems like overfitting. However, we are given a collection of tasks, the meta-training set $\mathcal{D}^{meta-train} = \{\mathcal{D}_t^{train}, \mathcal{D}_t^{test}\}_{t=1}^T$, on which we train a *meta-learner* which produces a predictor on every task $\{\mathcal{D}_t^{train}, \mathcal{D}_t^{test}\}$ and generalizes well on the meta-testing $\mathcal{D}^{meta-test} = \{\mathcal{D}_s^{train}, \mathcal{D}_s^{test}\}_{s=1}^S$.

Consider a $K$-class classification task, each training set $\mathcal{D}^{train}$ consists of $N/K$ labelled examples for each of $K$ classes. The meta-training task is often referred to as $N/K$-*shot*-$K$-*way*. Refer to Figure 1 for an example visualization of 2-shot-3-way few-shot *image* classification.

Meta learning approaches for few-shot learning often involve an embedding network, and a base learner (learning algorithm). The embedding network maps training samples to an embedding space. We explain in later subsections how the 3d reconstruction is connected to meta learning in these two aspects.

### 3.2 SINGLE IMAGE 3D RECONSTRUCTION

A watertight shape can be represented by an indicator (or occupancy) function $\mathcal{O} : \mathbb{R}^3 \to \{0, 1\}$. We define $\mathcal{O}(\mathbf{x}) = 1$ if $\mathbf{x} \in \mathbb{R}^3$ is inside the object, $\mathcal{O}(\mathbf{x}) = 0$ otherwise. We can sample a set of points in $\mathbb{R}^3$ and evaluate the indicator $\mathcal{O}$, then we have the labeled point set $\{\mathbf{x}_i, y_i\}_{i=1}^M$ where $y_i \in \{0, 1\}$. The number $M$ needs to be large enough to approximate the shape. In this way, we rewrite the target ground-truth as a point set. This strategy is also used by Mescheder et al. (2019) and Deng et al. (2019). Also see Figure 1 for an illustration.

The goal of single image 3D reconstruction is to convert an input image $\mathbf{I}$ to the indicator function $\mathcal{O}$. Previous work either directly learns $\mathcal{O}$ (Mescheder et al., 2019))or trains a network to predict an intermediate parametric representation (*e.g.* collection of convex primitives (Deng et al., 2019), half-spaces (Chen et al., 2019)). Different from any existing methods, our shape representation is to generate training data for a few-shot classification problem. In order to make the connection clear, we denote the ground-truth $\{\mathbf{x}_i, y_i\}_{i=1}^M$ as $\mathcal{D}^{test}$.

The training data of single image 3D reconstruction are a collection of images $\mathbf{I}_t$ and their corresponding shapes $\mathcal{D}_t^{test}$ which we denote as $\mathcal{D}^{meta-train} = \{\mathbf{I}_t, \mathcal{D}_t^{test}\}_{t=1}^T$. The goal is to learn a network which takes as input an image $\mathbf{I}$ and outputs a functional (predictor) $\mathcal{P}(\mathbf{x})$ which works on $\mathcal{D}^{test}$.

We summarize the notation and the mapping of few-shot classification to 3D shape reconstruction in Table 1. Using the proposed mapping, we need to find a *data generating* network (Fig. 2 left) to

Table 1: Symbols for few-shot classification and 3D reconstruction. Rows shown in blue are items 3D shape reconstruction has but few-shot classification does not.

| | Few-shot classification | 3D shape reconstruction |
|---|---|---|
| $\mathbf{I}$ | - | input images |
| $f$ | - | $\mathcal{D}^{train} = f(\mathbf{I})$ |
| $\mathcal{D}^{train}$ | $\{\mathbf{x}_i, y_i\}_{i=1}^N$ | - |
| $\mathcal{D}^{test}$ | $\{\mathbf{x}_i, y_i\}_{i=1}^M$ | $\{\mathbf{x}_i, y_i\}_{i=1}^M$ |
| $\mathbf{x}_i$ | images | points |
| $y_i$ | categories | inside/outside labels |
| predictor $\mathcal{P}(\cdot)$ | classifier | surface boundary |
| $\mathcal{D}^{meta-train}$ | $\{\mathcal{D}_t^{train}, \mathcal{D}_t^{test}\}_{t=1}^T$ | $\{\mathbf{I}_t, \mathcal{D}_t^{test}\}_{t=1}^T$ |

convert the input $\mathbf{I}$ to a set of labeled points $\mathcal{D}^{train} = \{\mathbf{x}_i, y_i\}_{i=1}^N$ (usually $N$ is far smaller than $M$). Then the $\mathcal{D}^{meta-train}$ can be rewritten as $\{\mathcal{D}_t^{train}, \mathcal{D}_t^{test}\}_{t=1}^T$. It can be seen that, this formulation has a high resemblance to few-shot learning. Also see Figure 1 for a visualization. As a result, we can leverage techniques from the literature of few-shot learning to jointly train the data generation and the classification components.

### 3.3 FORMULATION

Similar to few-shot learning, the problem can be written as a bi-level optimization. The inner optimization is to train the predictor $\mathcal{P}(\mathbf{x})$ to estimate the inside/outside label of a point,

$$\min_{\mathcal{P}} \mathbb{E}_{(\mathbf{x},y)\in\mathcal{D}^{train}} \left[\mathcal{L}\left(y, \mathcal{P}(\mathbf{x})\right)\right], \tag{1}$$

where $\mathcal{L}(\cdot, \cdot)$ is a loss function such as cross entropy. While in few-shot learning $\mathcal{D}^{train}$ is provided or sampled, here $\mathcal{D}^{train}$ is *generated* by a network $f$, $\mathcal{D}^{train} = f(\mathbf{I})$. To reconstruct the shape $(\mathbf{I}, \mathcal{D}^{test})$, the predictor $\mathcal{P}$ should work as an approximation of the indicator $\mathcal{O}$ and is expected to minimize the term,

$$\mathbb{E}_{(\mathbf{x},y)\in\mathcal{D}^{test}} \left[\mathcal{L}\left(y, \mathcal{P}(\mathbf{x})\right)\right]. \tag{2}$$

This process is done by a *base learner* (machine learning algorithm) in some meta learning methods (Bertinetto et al., 2019; Lee et al., 2019). The final objective across all shapes (tasks) is

$$\begin{aligned}
\min \ &\mathbb{E}_{(\mathbf{I},\mathcal{D}^{test})\in\mathcal{D}^{meta-train}} \left[\mathbb{E}_{(\mathbf{x},y)\in\mathcal{D}^{test}} \left[\mathcal{L}\left(y, \mathcal{P}(\mathbf{x})\right)\right]\right], \\
\text{s.t. } &\mathcal{P} = \min_{\mathcal{P}} \mathbb{E}_{(\mathbf{x},y)\in\mathcal{D}^{train}} \left[\mathcal{L}\left(y, \mathcal{P}(\mathbf{x})\right)\right], \quad \mathcal{D}^{train} = f(\mathbf{I}),
\end{aligned} \tag{3}$$

which is exactly the same as for meta learning algorithms if we remove the constraint $\mathcal{D}^{train} = f(\mathbf{I})$.

**Point Embedding.** In meta learning approaches for few-shot classification, an embedding network is used to map the training samples to an embedding space, $g(\mathbf{x}) = \mathbf{e}$, where $\mathbf{e}$ is the embedding vector of the input $\mathbf{x}$. We also migrate the idea to 3d shape representations, $g(\mathbf{x}|\mathbf{I}) = \mathbf{e}$, where the embedding network is also conditioned on the task input $\mathbf{I}$.

In later sections, we use $\mathbf{e}_i$ and $\mathbf{e}$ to denote the embeddings of point $\mathbf{x}_i$ and $\mathbf{x}$, respectively. In addition to the objective Eq equation 3, we add a regularizer,

$$w \cdot \mathbb{E}_{\mathbf{x}} \|\mathbf{e} - \mathbf{x}\|_2^2, \tag{4}$$

where $w$ is the weight for the regularizer. There are multiple solutions for the Embedding Network, thus we want to use the regularizer to shrink potential solutions, which is to find a similar embedding space to the original one. The $w$ is set to 0.01 in all experiments if not specified.

### 3.4 DIFFERENTIABLE LEARNER

The Eq. equation 1 is the inner loop of the final objective Eq. equation 3. Recent meta learning approaches use differentiable learners, *e.g.*, R2D2 (Bertinetto et al., 2019) uses Ridge Regression and MetaOptNet (Lee et al., 2019) uses Support Vector Machines (SVM). We describe both cases here. Different than the learner in R2D2 and MetaOptNet, we use kernelized algorithms.

**Ridge Regression.** Given a training set $\mathcal{D}^{train} = \{\mathbf{x}_i, y_i\}_{i=1}^N$, The kernel ridge regression (Murphy, 2012) is formulated as follows,

$$\underset{\boldsymbol{\alpha}}{\text{minimize}} \quad \frac{\lambda}{2}\boldsymbol{\alpha}^\intercal \mathbf{K}\boldsymbol{\alpha} + \frac{1}{2}\|\mathbf{y} - \mathbf{K}\boldsymbol{\alpha}\|_2^2 \quad , \tag{5}$$

where $\mathbf{K} \in \mathbb{R}^{N \times N}$ is the data kernel matrix in which each entry $\mathbf{K}_{i,j}$ is the Gaussian kernel $K(\mathbf{e}_i, \mathbf{e}_j) = \exp(-\|\mathbf{e}_i - \mathbf{e}_j\|_2^2/(2\sigma^2))$. The solution is given by an analytic form $\boldsymbol{\alpha} = (\mathbf{K} + \lambda\mathbf{I})^{-1}\mathbf{y}$. By using automatic differentiation of existing deep learning libraries, we can differentiate through $\boldsymbol{\alpha}$ with respect to each $\mathbf{x}_i$ in $\mathcal{D}^{train}$. We obtain the prediction for a query point $\mathbf{x}$ via

$$\text{RR}(\mathbf{e}; \mathcal{D}^{train}) = \sum_{i=1}^N \alpha_i K(\mathbf{e}_i, \mathbf{e}). \tag{6}$$

**SVM.** We use the dual form of kernel SVM,

$$\underset{\boldsymbol{\alpha}}{\text{minimize}} \quad \frac{1}{2}\sum_{j=1}^N \sum_{i=1}^N \alpha_i \alpha_j y_i y_j K(\mathbf{e}_i, \mathbf{e}_j) - \sum_{i=1}^N \alpha_i$$

$$\text{subject to} \quad \sum_{i=1}^N \alpha_i y_i = 0, \quad 0 \le \alpha_i \le C, \, i = 1, \dots, N, \tag{7}$$

where $K(\cdot, \cdot)$ is the Gaussian kernel. .

The discriminant function becomes,

$$\text{SVM}(\mathbf{e}; \mathcal{D}^{train}) = \sum_{i=1}^N \alpha_i y_i K(\mathbf{e}_i, \mathbf{e}) + b. \tag{8}$$

Using recent advances in differentiable optimization by Amos & Kolter (2017), the discriminant function is differentiable with respect to each $\mathbf{x}_i$ in $\mathcal{D}^{train}$.

While the definitions of both learners are given here, we only show results of ridge regression in our main paper. Additional results of SVM are provided in the supplementary material. In traditional machine learning, SVM is better than ridge regression in many aspects. However, we do not find SVM shows significant improvement over ridge regression in our experiments. The conclusion is also consistent with Lee et al. (2019).

### 3.5 INDICATOR APPROXIMATION

We clip the outputs of the ridge regression to the range $[0, 1]$ as an approximation $\hat{\mathcal{O}}_{RR}$ of the indicator $\mathcal{O}$. For the SVM, the predictor outputs a positive value if $\mathbf{x}$ is inside the shape otherwise a negative value. So we apply a sigmoid function to convert it to the range $[0, 1]$, $\hat{\mathcal{O}}_{SVM}(\mathbf{x}) = \text{Sigmoid}(\beta\text{SVM}(\mathbf{e}; \mathcal{D}^{train}))$, where $\beta$ is a learned scale. Then Eq. equation 2 is written as follows:

$$\mathbb{E}_{(\mathbf{x},y)\in\mathcal{D}^{test}} \left[ \left\| \hat{\mathcal{O}}(\mathbf{x}) - y \right\|_2^2 \right], \tag{9}$$

where the minimum squared error (MSE) loss is also used in CvxNet (Deng et al., 2019).

## 4 IMPLEMENTATION

**Base learner.** We use $\lambda = 0.005$ for ridge regression and $C = 1$ for SVM in all experiments. The parameter $\sigma$ in the kernel function $K(\cdot, \cdot)$ is learned during training and is shape-specific, i.e., each shape has its own $\sigma$.

Similar to recent works on instance segmentation (Liang et al., 2017; Kendall et al., 2018; Novotny et al., 2018; Zhang & Wonka, 2019), we also find that a simple $\mathbb{R}^3$ spatial embedding works well, i.e., $\mathbf{x} \in \mathbb{R}^3$ and $\boldsymbol{\sigma} \in \mathbb{R}^3$. Another reason for choosing $\mathbb{R}^3$ embedding is that we want to visualize the relationship between the original and embedding space in later sections.

Table 2: **Reconstruction results on ShapeNet.** We compare our results with Pixel2Mesh(P2M) (Wang et al., 2018), AtlasNet(AN) (Groueix et al., 2018), Occ-Net(ON) (Mescheder et al., 2019), SIF (Genova et al., 2019), CvxNet(CN) (Deng et al., 2019) and Hypernetwork(HN) (Littwin & Wolf, 2019). Best results are shown in bold.

| Categ. | IoU ↑ | | | | | | | Chamfer ↓ | | | | | | | | F-Score ↑ | | | | | | |
|---|---|---|---|---|---|---|---|---|---|---|---|---|---|---|---|---|---|---|---|---|---|---|
| | Ours | P2M | ON | ON† | SIF | CN | HN† | Ours | P2M | AN | ON | ON† | SIF | CN | HN† | Ours | AN | ON | ON† | SIF | CN | HN† |
| plane | **0.633** | 0.420 | 0.571 | 0.603 | 0.530 | 0.598 | 0.608 | 0.121 | 0.187 | 0.104 | 0.147 | 0.144 | 0.167 | **0.093** | 0.127 | **71.15** | 67.24 | 62.87 | 67.25 | 52.81 | 68.16 | 68.56 |
| bench | **0.524** | 0.323 | 0.485 | 0.486 | 0.333 | 0.461 | 0.501 | **0.132** | 0.201 | 0.138 | 0.155 | 0.148 | 0.261 | 0.133 | 0.146 | **69.44** | 54.50 | 56.91 | 64.80 | 37.31 | 54.64 | 65.43 |
| cabinet | 0.732 | 0.664 | 0.733 | 0.733 | 0.648 | 0.709 | **0.734** | **0.141** | 0.196 | 0.175 | 0.167 | 0.142 | 0.233 | 0.160 | 0.145 | **65.33** | 46.43 | 61.79 | 62.78 | 31.68 | 46.09 | 63.15 |
| car | **0.748** | 0.552 | 0.737 | 0.738 | 0.657 | 0.675 | 0.732 | 0.121 | 0.180 | 0.141 | 0.159 | 0.125 | 0.161 | **0.103** | 0.131 | **65.48** | 51.51 | 56.91 | 63.68 | 37.66 | 47.33 | 61.46 |
| chair | **0.532** | 0.396 | 0.501 | 0.515 | 0.389 | 0.491 | 0.517 | **0.205** | 0.265 | 0.209 | 0.228 | 0.217 | 0.380 | 0.337 | 0.230 | **48.83** | 38.89 | 42.41 | 46.38 | 26.90 | 38.49 | 45.56 |
| display | 0.553 | 0.490 | 0.471 | 0.540 | 0.491 | **0.576** | 0.542 | 0.215 | 0.239 | **0.198** | 0.278 | 0.213 | 0.401 | 0.223 | 0.217 | **46.96** | 42.79 | 38.96 | 44.55 | 27.22 | 40.69 | 44.42 |
| lamp | 0.383 | 0.323 | 0.371 | **0.395** | 0.260 | 0.311 | 0.389 | 0.391 | 0.308 | **0.305** | 0.479 | 0.404 | 1.096 | 0.795 | 0.441 | 42.99 | 33.04 | 38.35 | **43.41** | 20.59 | 31.41 | 41.81 |
| speaker | 0.658 | 0.599 | 0.647 | 0.651 | 0.577 | 0.620 | **0.661** | 0.247 | 0.285 | **0.245** | 0.300 | 0.261 | 0.554 | 0.462 | 0.261 | **46.86** | 35.75 | 42.48 | 44.05 | 22.42 | 29.45 | 45.33 |
| rifle | **0.540** | 0.402 | 0.474 | 0.496 | 0.463 | 0.515 | 0.508 | 0.111 | 0.164 | 0.115 | 0.141 | 0.129 | 0.193 | **0.106** | 0.120 | **69.40** | 64.22 | 56.52 | 64.34 | 53.20 | 63.74 | 65.92 |
| sofa | **0.707** | 0.613 | 0.680 | 0.692 | 0.606 | 0.677 | 0.692 | **0.155** | 0.212 | 0.177 | 0.194 | 0.167 | 0.272 | 0.164 | 0.163 | **56.40** | 43.46 | 48.62 | 53.01 | 30.94 | 42.11 | 53.13 |
| table | **0.551** | 0.395 | 0.506 | 0.534 | 0.372 | 0.473 | 0.531 | **0.171** | 0.218 | 0.190 | 0.189 | 0.179 | 0.454 | 0.358 | 0.189 | 65.25 | 44.93 | 58.49 | 63.67 | 30.78 | 48.10 | 63.35 |
| phone | **0.779** | 0.661 | 0.720 | 0.756 | 0.658 | 0.679 | 0.763 | 0.106 | 0.149 | 0.128 | 0.140 | 0.109 | 0.159 | **0.083** | 0.107 | **75.96** | 58.85 | 66.09 | 71.96 | 45.61 | 59.64 | 73.56 |
| vessel | **0.567** | 0.397 | 0.530 | 0.553 | 0.502 | 0.552 | 0.558 | 0.186 | 0.212 | **0.151** | 0.218 | 0.193 | 0.208 | 0.173 | 0.199 | **51.57** | 49.87 | 42.37 | 49.48 | 36.04 | 45.88 | 49.03 |
| mean | **0.608** | 0.480 | 0.571 | 0.592 | 0.499 | 0.567 | 0.595 | 0.177 | 0.217 | **0.175** | 0.215 | 0.187 | 0.349 | 0.245 | 0.190 | **59.66** | 48.58 | 51.75 | 56.87 | 34.86 | 47.36 | 56.98 |

† Re-implemeneted version.

**Networks.** Our framework is composed of three sub-networks: Feature Network, Point Generation Network and Embedding Network (see Fig. 2). For the Feature Network, we use EfficientNet-B1 (Tan & Le, 2019) to generate a $256$-dimensional feature vector. This architecture provides a good tradeoff between performance and simplicity. Both the Point Generation Network and Embedding Network are implemented with MLPs (see the supplementary material for the detailed architectures). The Point Generation Network outputs $\sigma$ and points $\{\mathbf{x}_i\}_{i=1}^{N}$ (half of which have $+1$ inside label and the other half have $-1$ outside label) where $N = 64$. The Embedding Network takes as input the concatenation of both the point $\mathbf{x}$ and the feature vector $\boldsymbol{\lambda}$. Instead of outputting $\mathbf{e}$ directly, we predict the offset $\mathbf{o} = \mathbf{e} - \mathbf{x}$ and apply the $\tanh$ activation to restrict the output to lie inside a bounding box. The training batch size is $32$. We use Adam (Kingma & Ba, 2014) with learning rate $2e - 4$ as our optimizer. The learning rate is decayed with a factor of $0.1$ after $500$ epochs.

**Data.** We perform single image 3d reconstruction on the ShapeNet (Chang et al., 2015) dataset. The rendered RGB images and data split are taken from (Choy et al., 2016). We sample 100k points uniformly from the shape bounding box as in OccNet (Mescheder et al., 2019) and also 100k "near-surface" points as in CvxNets (Deng et al., 2019) and SIF (Genova et al., 2019). Along with the corresponding inside/outside labels, we construct $\mathcal{D}^{test}$ for each shape offline to increase the training speed. At training time, 1024 points are drawn from the bounding box and 1024 "near-surface". This is the sampling strategy proposed by CvxNet.

## 5 RESULTS

**Evaluation metrics.** We use the volumetric IoU, the Chamfer-L1 distance and F-Score (Tatarchenko et al., 2019) for evaluation. Volumetric IoU is obtained from 100k uniformly sampled points. The Chamfer-L1 distance is estimated by randomly sampling 100k points from the ground-truth mesh and predicted mesh which is generated by Marching Cubes (Lorensen & Cline, 1987). F-Score is calculated with $d = 2\%$ of the side length of the reconstructed volume. Note that following the discussions by Tatarchenko et al. (2019), F-Score is a more robust and important metric for 3d reconstruction compared to IoU and Chamfer. All three metrics are used in CvxNet (Deng et al., 2019).

**Competing methods.** The list of competing methods includes Pixel2Mesh (Wang et al., 2018), AtlasNet (Groueix et al., 2018), SIF (Genova et al., 2019), OccNet (Mescheder et al., 2019), CvxNet (Deng et al., 2019) and Hypernetwork (Littwin & Wolf, 2019). Results are taken from these works except for (Littwin & Wolf, 2019) which we provide re-implemented results.

**Quantitative results.** We compare our method with a list of state-of-the-art methods quantitatively in Table 2. We improve the most important metric, F-score, from $51.75\%$ to $59.66\%$ compared to the previous state of the art OccNet (Mescheder et al., 2019). We also improve upon OccNet

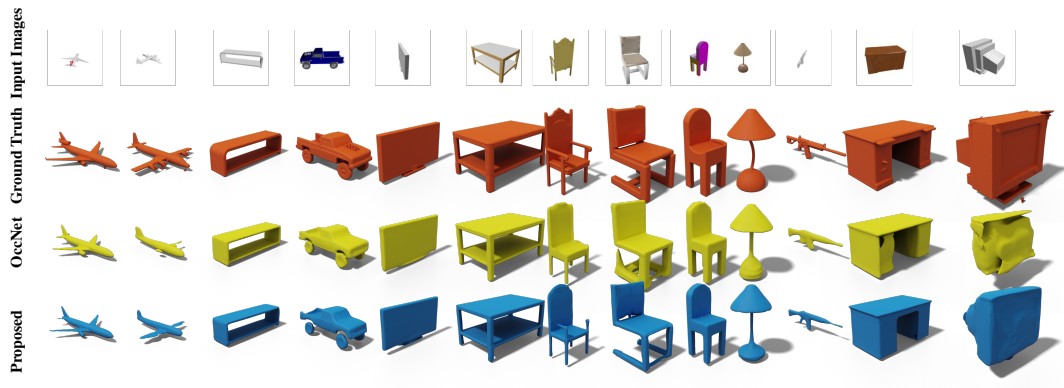

Figure 3: **Top:** Input images ($137 \times 137$) and ground-truth meshes shown in vermilion red. **Middle:** predicted meshes from (re-implemented) OccNet (Mescheder et al., 2019) shown in yellow. **Bottom:** predicted meshes from our model shown in blue.

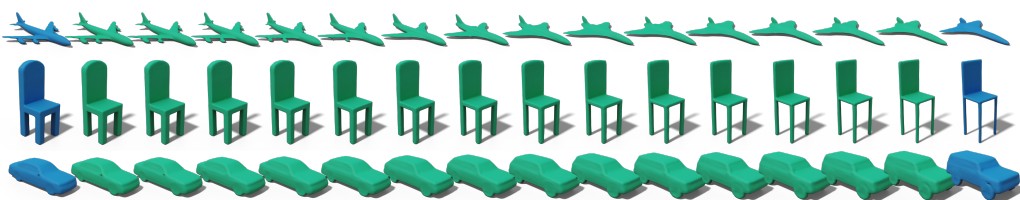

Figure 4: **Reconstruction with interpolated feature vectors.** Meshes in blue are original, green meshes are interpolated.

in the two other metrics. According to the L1-Chamfer metric, AtlasNet (Groueix et al., 2018) has a slight edge, but we would like to reiterate that this metric is less important and we list it mainly for completeness. Besides that, we re-implemented OccNet (Mescheder et al., 2019) using our embedding network and Hypernetwork (Littwin & Wolf, 2019) under our training setup. For OccNet, the re-implemented version has better results than the original one. For Hypernetwork, we carefully choose the hyper-parameters as the paper suggested. Overall, we can observe that our method has the best results compared to all other methods.

**Qualitative results.** We show qualitative reconstruction results in Fig. 3. We compare our results with re-implemented OccNet. We also show shape interpolation results in Fig. 4. Interpolation is done by acquiring two features $\boldsymbol{\lambda}_1$ and $\boldsymbol{\lambda}_2$ from two different images. Then we reconstruct meshes while linearly interpolating the two features.

**Ablation study.** We show the metrics under different hyper-parameter choices in Table 3, including different $N$ and $w$. We find that generally smaller $w$ and larger $N$ gives better results. However when $w$ is small enough ($w = 0.01, 0.1$), the metrics are very close no matter how large $N$ is.

**Visualization of embedding space.** In Fig. 5, we first show $\mathcal{D}^{train}$ along with the corresponding meshes. Since our embedding space is also $\mathbb{R}^3$, we can visualize the embedding space. When the number of points $N$ in $\mathcal{D}^{train}$ is small, the embedded shape looks more like an ellipsoid. In this case, the Embedding Network maps inside points into an ellipsoid-like shape. When $N$ is getting larger, the embedded mesh has a shape more similar to the original one. Thus, the Embedding Network only shift points by a small distance. This shows how the Embedding Network and the Point Generation Network collaborate. Sometimes the Embedding Network does most of the work when points are difficult to classify, while sometimes the Point Generation Network needs to generate *structured* point sets. In Fig. 6, we show more examples of $\mathcal{D}^{train}$ given $N = 64$ and $w = 0.01$.

**Relation to hypernetworks.** The approach of hypernetworks (Ha et al., 2016), uses a (hyper-)network to generate parameters of another network. Our method has a relation to hypernetworks. The final (decision) surface is decided by $\mathbf{e}_i = g(\mathbf{x}_i), i = 1, \ldots, N$ where $\mathbf{x}_i$ is a point in $\mathcal{D}^{train}$. According to Eq. equation 6, the output is $\sum_{i=1}^{N} \alpha_i K(g(\mathbf{x}_i), g(\mathbf{x}))$. The points $\mathcal{D}^{train} = \{\mathbf{x}_i, y_i\}_{i=1}^{N}$ is obtained by the Point Generation Network, which we can treat as a *hypernetwork*, and the points are *generated parameters*. Thus our method can also be interpreted as a hypernetwork. Littwin & Wolf (2019) proposes an approach for shape reconstruction with hypernetworks but with a very high-dimensional latent vector (1024) and generated parameters (3394), while we only use a 256 dimensional latent vector and $3 \times 64 = 192$ parameters. We still have better results according to Table 2.

Table 3: **Ablation study.** We show (a) IoU $\uparrow$, (b) Chamfer $\downarrow$ and (c) F-Score $\uparrow$ for different $N$ and $w$.

(a) IoU $\uparrow$

| IoU | 256 | 128 | 64 | 32 | 16 | 8 |
|---|---|---|---|---|---|---|
| $w = 10$ | 0.582 | 0.578 | 0.587 | 0.583 | 0.568 | 0.553 |
| $w = 5$ | 0.599 | 0.600 | 0.595 | 0.588 | 0.580 | 0.574 |
| $w = 2$ | 0.600 | 0.604 | 0.600 | 0.595 | 0.589 | 0.584 |
| $w = 1$ | 0.603 | 0.607 | 0.604 | 0.596 | 0.595 | 0.592 |
| $w = 0.1$ | 0.610 | 0.606 | 0.608 | 0.609 | 0.611 | 0.607 |
| $w = 0.01$ | 0.606 | 0.610 | 0.608 | 0.608 | 0.613 | 0.610 |
| $w = 0$ | 0.607 | 0.606 | 0.608 | 0.605 | 0.609 | 0.611 |

(b) Chamfer $\downarrow$

| Chamfer | 256 | 128 | 64 | 32 | 16 | 8 |
|---|---|---|---|---|---|---|
| $w = 10$ | 0.208 | 0.224 | 0.235 | 0.273 | 0.282 | 0.317 |
| $w = 5$ | 0.198 | 0.212 | 0.226 | 0.242 | 0.270 | 0.285 |
| $w = 2$ | 0.189 | 0.196 | 0.210 | 0.223 | 0.230 | 0.252 |
| $w = 1$ | 0.186 | 0.191 | 0.199 | 0.211 | 0.216 | 0.220 |
| $w = 0.1$ | 0.180 | 0.179 | 0.179 | 0.176 | 0.175 | 0.178 |
| $w = 0.01$ | 0.182 | 0.175 | 0.177 | 0.175 | 0.171 | 0.173 |
| $w = 0$ | 0.178 | 0.182 | 0.176 | 0.178 | 0.173 | 0.170 |

(c) F-Score $\uparrow$

| F-Score | 256 | 128 | 64 | 32 | 16 | 8 |
|---|---|---|---|---|---|---|
| $w = 10$ | 0.542 | 0.529 | 0.551 | 0.544 | 0.521 | 0.494 |
| $w = 5$ | 0.575 | 0.573 | 0.565 | 0.554 | 0.544 | 0.533 |
| $w = 2$ | 0.579 | 0.582 | 0.578 | 0.567 | 0.561 | 0.552 |
| $w = 1$ | 0.585 | 0.586 | 0.585 | 0.575 | 0.571 | 0.570 |
| $w = 0.1$ | 0.597 | 0.591 | 0.598 | 0.597 | 0.599 | 0.593 |
| $w = 0.01$ | 0.593 | 0.599 | 0.597 | 0.596 | 0.600 | 0.598 |
| $w = 0$ | 0.594 | 0.591 | 0.591 | 0.590 | 0.597 | 0.601 |

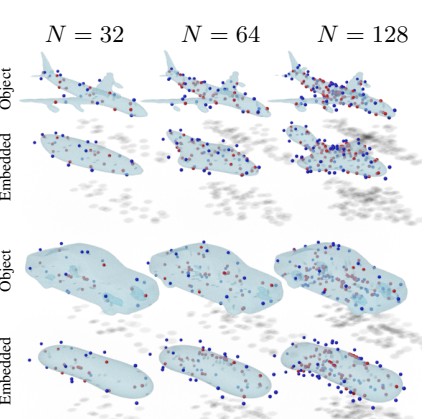

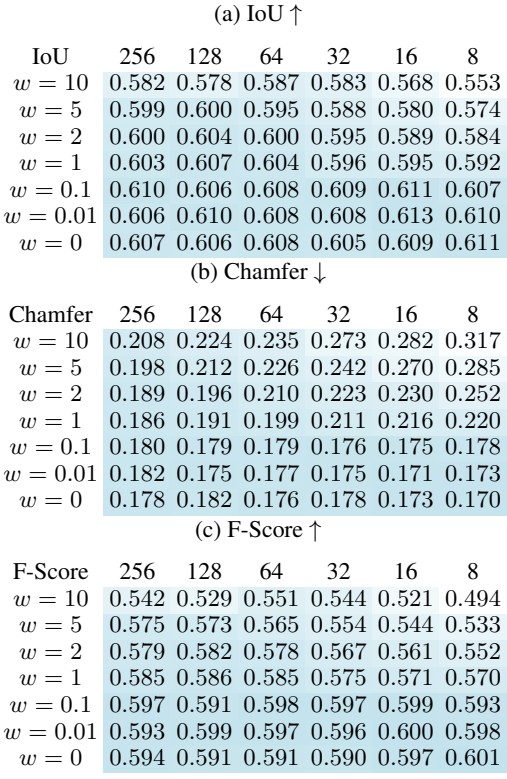

Figure 5: We show $\mathcal{D}^{train}$ and $\mathcal{D}^{train}$ in embedding space given $N = 32, 64, 128$ and $w = 0.01$. Positive points are shown in red and negatives points in blue. The shapes in both spaces are shown as transparent surfaces.

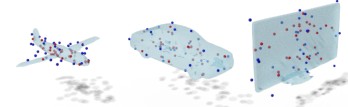

Figure 6: Visualizations of $\mathcal{D}^{train}$ when $N = 64$ and $w = 0.01$. Positive points in red and negative points in blue.

## 6 CONCLUSION

In this paper, we presented a shape representation for deep neural networks. Training data generating networks establish a connection between few-shot learning and shape representation by converting an image of a shape into a collection of points as training set for a supervised task. Training can be solved with meta-learning approaches for few-shot learning. While our solution is inspired by few-shot learning, it is different in: 1) our training datasets are generated by a separate network and not given directly; 2) our embedding network is conditioned on the task but traditional few-shot learning employs unconditional embedding networks; 3) our test dataset is generated by sampling and not directly given. The experiments are evaluated on a single image 3D reconstruction dataset and improve over the SOTA.

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

## A  APPENDIX

### A.1  NETWORKS

We show the architecture of the embedding network and the points generator in Fig. 7. The embedding network is similar to DeepSDF (Park et al., 2019).

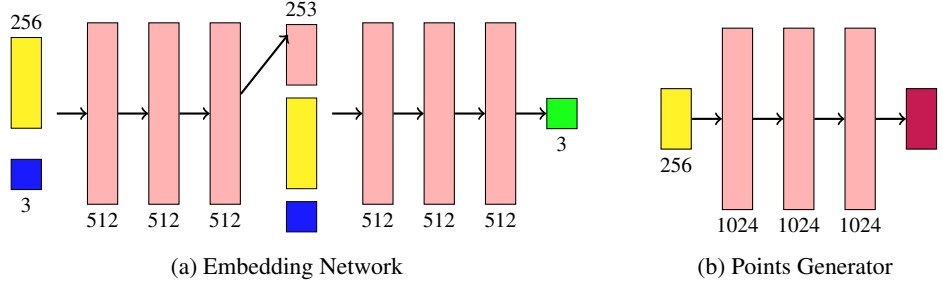

(a) Embedding Network          (b) Points Generator

Figure 7: **Networks.** We use fully connected layers in both networks. In (a), the input is the concatenation of feature $\boldsymbol{\lambda}$ (yellow) and point $\mathbf{x}$ (blue). In (b), the input is the feature $\boldsymbol{\lambda}$ (yellow), and the output is the coordinates of points $\mathbf{X} \in \mathbf{R}^{N \times 3}$ (red).

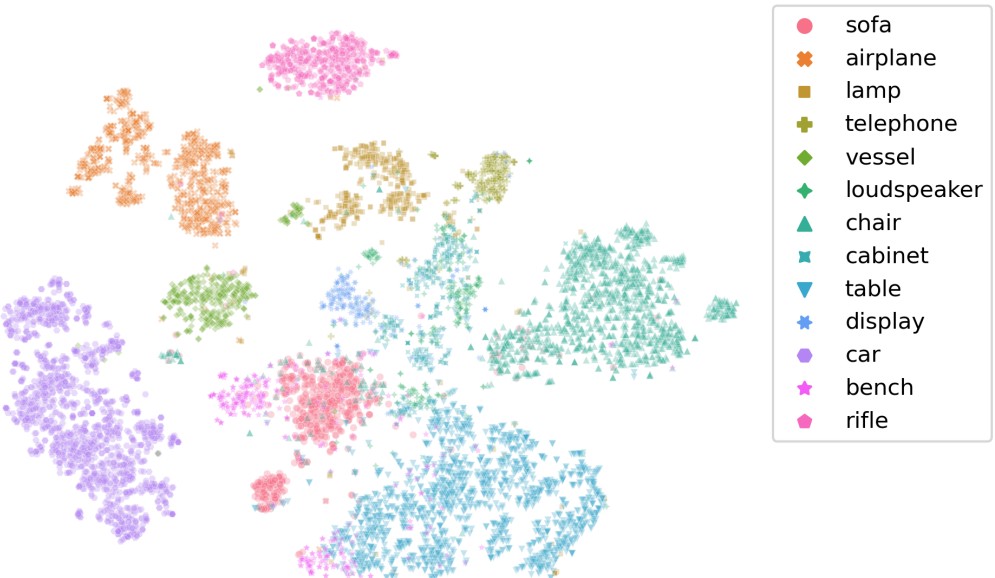

Figure 8: **T-SNE visualization** using the extracted features of the shapes.

### A.2 VISUALIZATION OF FEATURE SPACE

We visualize the feature space in Fig. 8.

### A.3 STATISTICS OF METRICS

Here we show more detailed statistics in Fig. 9. Comparing to IoU and F-score, most of the values of Chamfer concentrate around the mean value and rare values very far from the mean. The "peakedness" property of Chamfer implies it is unstable to a certain extent. The conclusion is also consistent with Tatarchenko et al. (2019).

### A.4 SHAPE INTERPOLATION

In Fig. 10, we show reconstructed shapes with bilinear interpolation. Given 4 shape latent feature vectors $\boldsymbol{\lambda}_{0,0}$, $\boldsymbol{\lambda}_{0,1}$, $\boldsymbol{\lambda}_{1,0}$ and $\boldsymbol{\lambda}_{1,1}$, we show interpolated shapes with the bilinear interpolation equation

$$\boldsymbol{\lambda}_{v,h} = (1-v)(1-h)\boldsymbol{\lambda}_{0,0} + (1-v)h\boldsymbol{\lambda}_{0,1} + v(1-h)\boldsymbol{\lambda}_{1,0} + vh\boldsymbol{\lambda}_{1,1}$$

where $v \in [0,1]$ and $h \in [0,1]$

### A.5 EFFECTS OF $w$

In the main paper, we have a regularizer to force the embedding space to be similar to the original space,

$$w \cdot \mathbb{E}_{\mathbf{x}} \left\| \mathbf{e} - \mathbf{x} \right\|_2^2. \tag{10}$$

When $w$ is very large, the embedding space is almost the same as the original space, especially when $N$ is large. The visualization is shown in Fig. 11. We set $w = 10$. This is much higher than $w = 0.01$ recommended in the paper. In this case, we expect the embedding network to do less work and as reported for $w = 10$ the method mainly works for a large number of points. We can observe that indeed the embedding space (bottom) is similar to the original space (top) especially when $N$ is large. Positive points are shown in red and negative points in blue. **Top row:** meshes are reconstructed with our regular pipeline. Generated points are fed into the Embedding Network $g$ first. Visualized points are in the original space. **Middle row:** meshes are reconstructed by removing

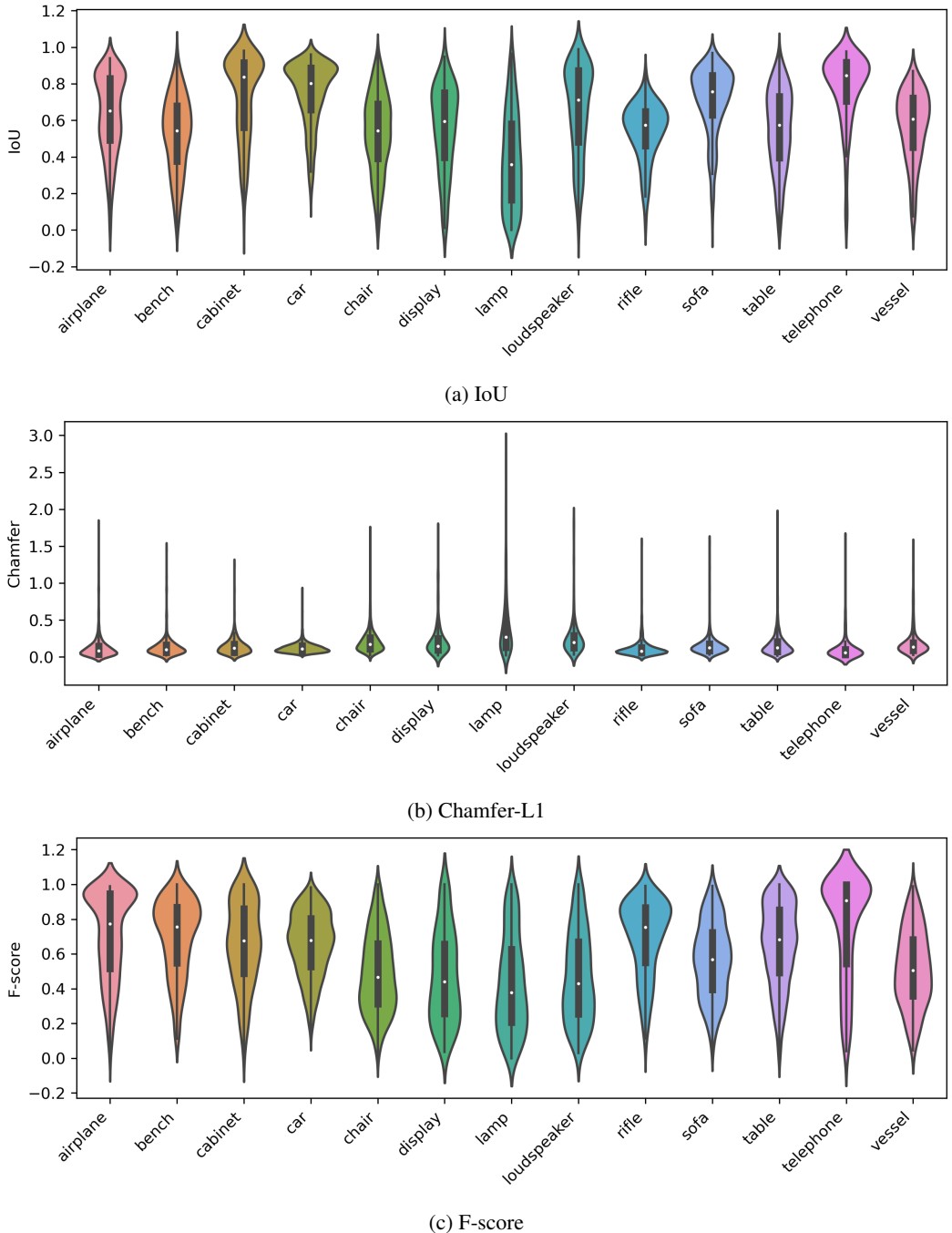

(a) IoU

(b) Chamfer-L1

(c) F-score

Figure 9: Statistics of metric.

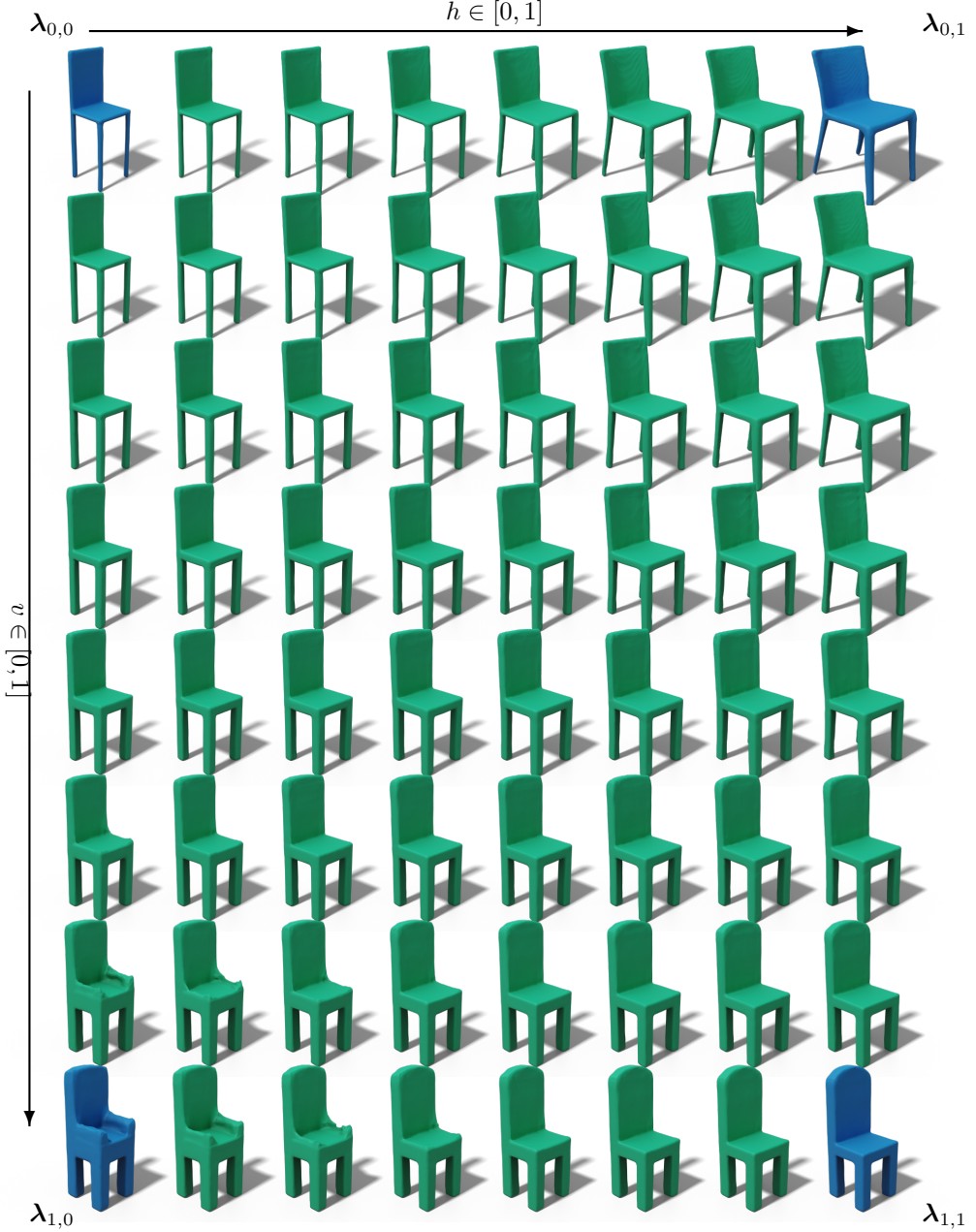

Figure 10: **Reconstruction with bilinear interpolated feature vectors.** Meshes in blue are original, green meshes are interpolated. The bilinear equation is $\boldsymbol{\lambda}_{v,h} = (1-v)(1-h)\boldsymbol{\lambda}_{0,0} + (1-v)h\boldsymbol{\lambda}_{0,1} + v(1-h)\boldsymbol{\lambda}_{1,0} + vh\boldsymbol{\lambda}_{1,1}$

| Category | IoU ↑ | | Chamfer ↓ | | F-Score ↑ | |
|---|---|---|---|---|---|---|
| | SVM | RR | SVM | RR | SVM | RR |
| airplane | **0.639** | 0.633 | **0.117** | 0.121 | **71.87** | 71.15 |
| bench | 0.522 | **0.524** | 0.144 | **0.132** | 67.88 | **69.44** |
| cabinet | 0.731 | **0.732** | **0.139** | 0.141 | 65.15 | **65.33** |
| car | 0.745 | **0.748** | **0.119** | 0.121 | **65.77** | 65.48 |
| chair | 0.526 | **0.532** | 0.229 | **0.205** | 47.73 | **48.83** |
| display | 0.552 | **0.553** | **0.213** | 0.215 | **47.88** | 46.96 |
| lamp | **0.388** | 0.383 | 0.398 | **0.391** | 41.94 | **42.99** |
| loudspeaker | **0.662** | 0.658 | 0.252 | **0.247** | **46.95** | 46.86 |
| rifle | 0.536 | **0.540** | 0.112 | **0.111** | 69.18 | **69.40** |
| sofa | 0.700 | **0.707** | 0.158 | **0.155** | 55.40 | **56.40** |
| table | 0.538 | **0.551** | 0.193 | **0.171** | 64.03 | **65.25** |
| telephone | 0.776 | **0.779** | **0.101** | 0.106 | **76.60** | 75.96 |
| vessel | 0.563 | **0.567** | **0.177** | 0.186 | 51.38 | **51.57** |
| mean | 0.606 | **0.608** | 0.181 | **0.177** | 59.36 | **59.66** |

Table 4: Results of different base learners: SVM vs Ridge Regression (RR).

the embedding network *without re-training*. This isn't expected to work well. Generated points are *directly* used in generating classification boundaries. Visualized points are in the original space. We can see that even without the embedding network the results are surprisingly reasonable for a large number of points. **Bottom row:** meshes in the embedding space. Visualized points are in the embedding space. We can observe that the meshes look similar to the correct chair, especially for large $N$. This also validates that the embedding network does not have a large influence on the result for large $w$. If the regularizing weight $w$ is set to 0, the shape in embedding space will be more ellipsoid like.

Recall that, with the kernel ridge regression as our base learner, the final shape (decision) surface is decided by,

$$\sum_{i=1}^{N} \alpha_i K(g(\mathbf{x}_i), g(\mathbf{x})).$$

If the original and embedding space are close, we can even reconstruct meshes without the Embedding Network $g$,

$$\sum_{i=1}^{N} \alpha_i K(\mathbf{x}_i, \mathbf{x}).$$

The results are shown in Fig. 11.

### A.6 THE CHOICE OF BASE LEARNERS

In Table 4, we show results of different base learners, SVM and ridge regression. The difference between the two methods is very small.

### A.7 RESULTS ON REAL WORLD IMAGES

Note that our model is trained on a synthetic dataset (ShapeNet). Here we test how our model generalizes to real world datasets. Without retraining and fine-tuning, we apply our trained model to Stanford Online Products. The qualitative results are shown in Fig. 12. We believe that the results are good, but there is no ground truth data for evaluation.

### A.8 INTER-CATEGORIES INTERPOLATION

Similar to Fig. 4, we also show inter-categories interpolation in Fig. 13. We take samples from two different categories, and reconstruct shapes with interpolated latents. Although the interpo-

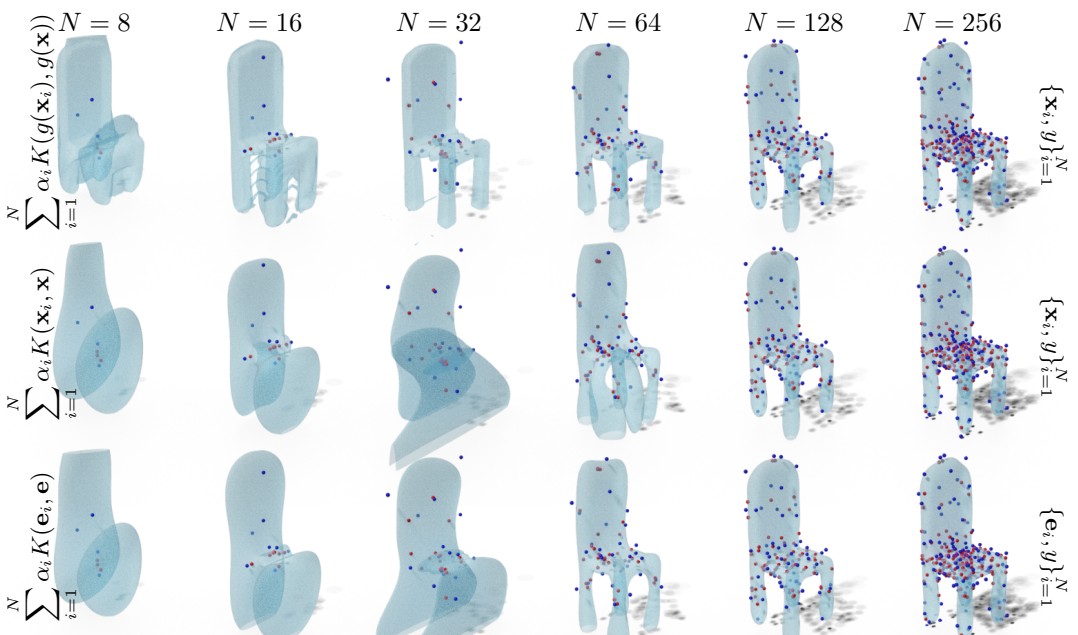

Figure 11: **Effects of large** $w$**.**

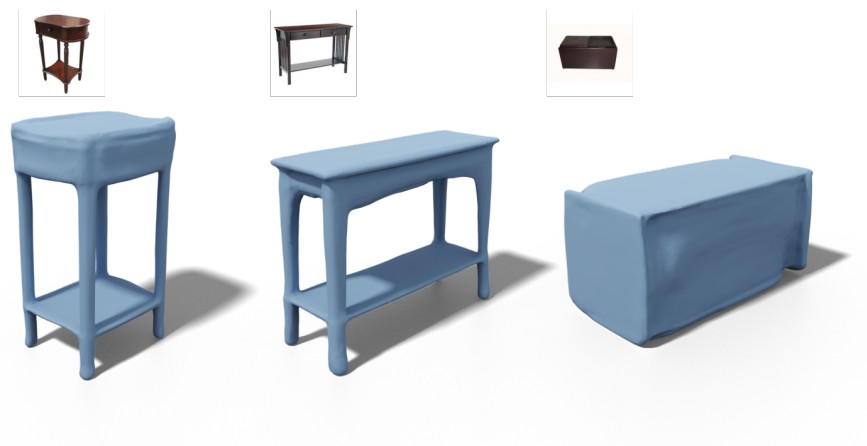

Figure 12: **Visual results for Stanford Online Products.** The input images are shown in top row. They are resized to $137 \times 137$ to fit the input requirement of our model.

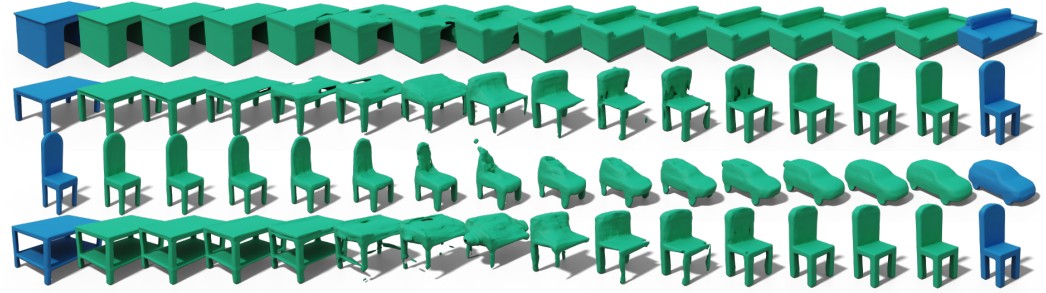

Figure 13: **Reconstruction with interpolated feature vectors.** Meshes in blue are original, green meshes are interpolated.

lated shapes are not as high quality than the originals, they still show some characteristics of both categories, *e.g.*, sofas which are as tall as tables, and cars with chair-leg-like tires.

### A.9 TRAINING AND INFERENCE

We describe the detailed training process in Algorithm 1. This algorithm can be viewed along with Fig. 2. We also show the full process how we generate a triangular mesh given an input image in Algorithm 2.

---

**Algorithm 1** Training

---

**for** number of training iterations **do**
    • Sample a minibatch of $m$ training sample pairs $\{\mathbf{I}_b, \mathbf{M}_b\}_{b=1}^B$, where $\mathbf{I}_b$ is an image and $\mathbf{M}_b$ is the target ground-truth mesh.
    • Get the latent representations for images with a feature backbone

$$\text{Feature}\left(\mathbf{I} \in \mathbb{R}^{B \times 3 \times H \times W}\right) = \mathbf{L} \in \mathbb{R}^{B \times 256},$$

    • Get the labeled point sets (the first half are marked as positive and the other half are marked as negative)

$$\text{PointGen}(\mathbf{L}) = \mathbf{X} \in \mathbb{R}^{B \times N \times 3}, \qquad \mathbf{y} \in \mathbb{R}^{B \times N}$$

    • Embed the point sets $\mathbf{X}$ to another space

$$\text{Embedding}(\mathbf{X}) = \mathbf{E} \in \mathbb{R}^{B \times N \times 3},$$

    • Optimize differentiable base learner $P(\cdot|\boldsymbol{\alpha})$ which defines binary classification boundaries,

$$\text{OptBaseLearner}(P, \mathbf{E}, \mathbf{y}) = \boldsymbol{\alpha} \in \mathbb{R}^{B \times N},$$

    • Sample labeled point sets from ground-truth meshes $\{\mathbf{M}_i\}_{b=1}^B$,

$$\mathbf{X}_t \in \mathbb{R}^{B \times M \times 3}, \qquad \mathbf{y}_t \in \mathbb{R}^{B \times M},$$

    • Convert $\mathbf{X}_t$ to the embedding space

$$\text{Embedding}(\mathbf{X}_t) = \mathbf{E}_t \in \mathbb{R}^{B \times N \times 3},$$

    • Predict labels by using base learner $P(\cdot|\boldsymbol{\alpha})$

$$P(\mathbf{E}_t|\boldsymbol{\alpha}) = \bar{\mathbf{y}}_t,$$

    • Update $\text{Feature}(\cdot)$, $\text{PointGen}(\cdot)$ and $\text{Embedding}(\cdot)$ by descending stochastic gradients

$$\nabla\text{MSE}(\bar{\mathbf{y}}_t, \mathbf{y}_t),$$

**end for**

---

---

**Algorithm 2** Inference

---

• Prepare an input image $\mathbf{I} \in \mathbb{R}^{3 \times H \times W}$,
• Get the latent representations for the image

$$\text{Feature}(\mathbf{I}) = \mathbf{L} \in \mathbb{R}^{256},$$

• Get the labeled point sets (the first half are marked as postive and the other half are marked as negative)

$$\text{PointGen}(\mathbf{L}) = \mathbf{X} \in \mathbb{R}^{N \times 3}, \qquad \mathbf{y} \in \mathbb{R}^{N}$$

• Embed points to the embedding space

$$\text{Embedding}(\mathbf{X}) = \mathbf{E} \in \mathbb{R}^{N \times 3},$$

• Optimize the differentiable base learner $P(\cdot | \boldsymbol{\alpha})$,

$$\text{OptBaseLearner}(P, \mathbf{E}, \mathbf{y}) = \boldsymbol{\alpha} \in \mathbb{R}^{N},$$

• Prepare a point set on a 3D grid with arbitrary resolution $G \times G \times G$,

$$\mathbf{X}_t \in \mathbb{R}^{G^3 \times 3}$$

• Convert $\mathbf{X}_t$ to embedding space

$$\text{Embedding}(\mathbf{X}_t) = \mathbf{E}_t \in \mathbb{R}^{G^3 \times 3},$$

• Predict labels by using base learner $P(\cdot | \boldsymbol{\alpha})$

$$P(\mathbf{E}_t | \boldsymbol{\alpha}) = \bar{\mathbf{y}}_t \in \mathbb{R}^{G \times G \times G},$$

• Apply marching cubes algorithm to $\bar{\mathbf{y}}_t$,

$$\text{MarchingCubes}(\bar{\mathbf{y}}_t) = (\mathbf{V}, \mathbf{F}).$$

---

Table 5: **Different backbones. ENB1**: EfficientNet-B1. **RN34**: ResNet-34. **RN50**: Resnet-50. **DN121**: DenseNet-121.

| Categ. | IoU ↑ | | | | Chamfer ↓ | | | | F-Score ↑ | | | |
|---|---|---|---|---|---|---|---|---|---|---|---|---|
| | ENB1 | RN34 | RN50 | DN121 | ENB1 | RN34 | RN50 | DN121 | ENB1 | RN34 | RN50 | DN121 |
| plane | **0.633** | 0.616 | 0.623 | 0.612 | **0.121** | 0.129 | 0.131 | 0.142 | **71.15** | 69.15 | 69.70 | 67.77 |
| bench | **0.524** | 0.506 | 0.500 | 0.508 | **0.132** | 0.135 | 0.152 | 0.145 | **69.44** | 67.24 | 65.69 | 66.32 |
| cabinet | 0.732 | 0.728 | 0.733 | **0.742** | 0.141 | 0.144 | 0.141 | **0.136** | 65.33 | 63.44 | 63.78 | **65.58** |
| car | **0.748** | 0.746 | 0.744 | 0.744 | **0.121** | 0.122 | 0.123 | 0.123 | **65.48** | 64.86 | 63.98 | 64.23 |
| chair | **0.532** | 0.520 | 0.521 | **0.532** | **0.205** | 0.213 | 0.219 | 0.216 | **48.83** | 47.33 | 47.14 | 48.11 |
| display | 0.553 | 0.549 | 0.534 | **0.554** | 0.215 | **0.209** | 0.222 | 0.210 | **46.96** | 46.45 | 43.83 | 46.85 |
| lamp | 0.383 | 0.378 | 0.370 | **0.386** | **0.391** | 0.399 | 0.434 | 0.410 | **42.99** | 41.38 | 39.49 | 41.81 |
| speaker | 0.658 | 0.647 | 0.647 | **0.659** | **0.247** | 0.262 | 0.265 | 0.252 | **46.86** | 44.47 | 43.77 | 46.52 |
| rifle | **0.540** | 0.521 | 0.515 | 0.504 | **0.111** | 0.124 | 0.118 | 0.124 | **69.40** | 67.19 | 66.47 | 65.09 |
| sofa | **0.707** | 0.691 | 0.694 | 0.702 | **0.155** | 0.164 | 0.166 | **0.155** | **56.40** | 54.32 | 54.18 | 55.62 |
| table | **0.551** | 0.535 | 0.532 | 0.550 | **0.171** | 0.177 | 0.200 | 0.176 | **65.25** | 63.92 | 63.26 | 65.22 |
| phone | **0.779** | 0.770 | 0.769 | 0.770 | 0.106 | **0.105** | 0.107 | **0.105** | **75.96** | 75.22 | 73.06 | 75.30 |
| vessel | **0.567** | 0.562 | 0.555 | 0.570 | 0.186 | **0.185** | 0.206 | 0.192 | **51.57** | 50.27 | 48.74 | 50.68 |
| mean | **0.608** | 0.598 | 0.595 | 0.602 | **0.177** | 0.182 | 0.191 | 0.183 | **59.66** | 58.10 | 57.16 | 58.39 |

Table 6: **Different embeddings. Spa**: (3D) spatial embeddings. **Other columns**: trained with linear ridge regression of different embedding dimensions (32, 64 and 128).

| Categ. | IoU ↑ | | | | Chamfer ↓ | | | | F-Score ↑ | | | |
|---|---|---|---|---|---|---|---|---|---|---|---|---|
| | Spa | 32D | 64D | 128D | Spa | 32D | 64D | 128D | Spa | 32D | 64D | 128D |
| plane | **0.637** | 0.627 | 0.630 | 0.624 | **0.119** | 0.125 | 0.132 | 0.123 | **71.54** | 70.67 | 70.91 | 70.30 |
| bench | 0.519 | 0.530 | **0.532** | 0.516 | 0.138 | 0.135 | 0.138 | **0.133** | 68.42 | 68.89 | **69.57** | 68.23 |
| cabinet | **0.744** | 0.735 | 0.739 | 0.736 | 0.135 | 0.136 | 0.138 | **0.134** | 66.54 | 66.92 | **67.13** | 66.77 |
| car | 0.747 | 0.746 | **0.749** | 0.746 | 0.119 | 0.121 | **0.118** | 0.119 | 65.51 | 65.53 | **65.98** | 65.50 |
| chair | 0.527 | 0.536 | **0.542** | 0.532 | 0.223 | 0.233 | 0.210 | **0.207** | 47.89 | 49.05 | **49.93** | 48.88 |
| display | **0.572** | 0.562 | 0.570 | 0.564 | 0.201 | 0.201 | **0.196** | 0.201 | 48.58 | 48.00 | **48.73** | 47.76 |
| lamp | 0.387 | 0.397 | **0.417** | 0.408 | 0.398 | 0.479 | 0.456 | **0.363** | 42.57 | 43.78 | **45.18** | 44.29 |
| speaker | 0.657 | **0.666** | **0.666** | 0.660 | 0.259 | 0.244 | **0.233** | 0.242 | 45.83 | 48.19 | **48.36** | 48.21 |
| rifle | 0.537 | 0.540 | **0.545** | 0.537 | 0.113 | 0.116 | 0.113 | **0.109** | 69.33 | 69.25 | **69.81** | 68.77 |
| sofa | 0.705 | 0.706 | **0.707** | **0.707** | 0.157 | 0.153 | 0.153 | **0.152** | 55.90 | 56.49 | **56.73** | 56.53 |
| table | 0.549 | 0.550 | **0.553** | 0.548 | 0.183 | 0.177 | 0.172 | **0.169** | 65.08 | 65.77 | **66.16** | 64.96 |
| phone | 0.779 | 0.780 | **0.786** | 0.777 | 0.107 | 0.100 | **0.094** | 0.097 | **76.95** | 76.83 | 76.29 | 76.06 |
| vessel | 0.575 | 0.572 | **0.577** | 0.569 | 0.189 | 0.199 | 0.189 | **0.183** | 51.75 | **52.10** | 52.03 | 51.84 |
| mean | 0.610 | 0.611 | **0.616** | 0.610 | 0.180 | 0.186 | 0.180 | **0.172** | 59.68 | 60.11 | **60.52** | 59.85 |

## A.10 CHOICE OF BACKBONES

In Table 5, we show the performance of our method with different backbones. We keep all of the setup the same except for the feature backbone.

## A.11 EMBEDDING DIMENSIONS

In Table 6, we show the performance of our method with different embeddings. Note that different from Table 5, the results shown here are trained with $N = 256$.

