# OpenReview forum: "Training Data Generating Networks: Shape Reconstruction via Bi-level Optimization"
_ICLR.cc/2022/Conference — ICLR 2022 Poster_

### Official Review · Reviewer_7UAm · 2021-11-01

**Correctness:** 3
**Technical Novelty And Significance:** 3
**Empirical Novelty And Significance:** 3
**Recommendation:** 8
**Confidence:** 4

**Main Review:**

Please Bist both the strengths and weaknesses of the paper. When discussing weakness, please provide concrete. actionable feedback on the paper.

To avoid fully sample points and its distance value to the surface over the 3D space in training, the paper proposes a meta learning approach. It is a novel idea in computer graphics because different from datasets in computer vision or natural language processing, 3D shape dataset is far smaller. However, I still have several concerns.

- The proposed architecture uses a point generation network to generate training data for few shot learning. I am wondering why not directly sample a few points on and near the shape surface to train the few shot network?

- Predicted points are not guaranteed to be exactly those points you have sampled in the offline setting. So how do you get the occupancy value at these points in the training process?

- Even though the author gives the reason for R^3 embedding space, I am wondering if how the embedding space dimension will influence the results.

**Summary Of The Paper:**

The paper introduce a few shot learning method for single view reconstruction task. Different from traditional few shot learning methods, data used for training are generated by a network which takes an image as input. Both qualitative and quantitative reconstruction results show that this method gives a better performance than current SOTA methods at less training data(occupancy or distance field ground truth).


**Summary Of The Review:**

I am generally positive about this paper since it proposes a novel method to single view reconstruction using few shot learning. But it needs a more detailed illustration about the training process and more evaluations on the design choice. If the authors can resolve those issues I mentioned above, I am willing to give a higher score.

---

> ### Author Response · Authors · 2021-11-20
> **Response to Reviewer 7UAm**
>
> We thank the reviewer for the feedback.
>
> ### Sampling on and near shape surfaces to train the network.
> The generated point set contains representative points for the input shapes. We design the method by making the point set to be generated from input images. It is possible to pick some labelled points on and near ground-truth meshes’ surfaces when training. But for inference, the ground-truth meshes are not known and we are unable to perform such a sampling. Since we are sampling a point set from an input image there would be a component missing if we only have the ability to sample from surfaces. Also, the point sets we learn to select from an input image are more efficient shape representations than randomly sampled points on and near the surface.
>
> ### How to get labels for predicted point sets.
> The predicted point sets are obtained from the image latents by the PointGenNetwork. The occupancy value (labels) are not directly output by the network. The PointGenNetwork outputs N points without labels. Half of this point set are with positive labels (inside) and the other half are negative (outside). The label is therefore given implicitly by the order of the points. We do as the reviewer suggested to include the full training process in the revised appendix (A.9, Algorithm 1).
>
> ### Ablation study about embedding dimensions.
> We tested different embedding dimensions as the reviewer suggested. The results can be seen in the revised appendix (A.11, Table 6). Higher embedding dimensions require larger GPU memory. For now we could not train 256D embeddings. Also please see answers to **reviewer DXHG** and **reviewer ABfL**.

---

### Official Review · Reviewer_DXHG · 2021-11-03

**Correctness:** 3
**Technical Novelty And Significance:** 3
**Empirical Novelty And Significance:** 3
**Recommendation:** 6
**Confidence:** 3

**Main Review:**

Strengths of the paper:
- The paper is clearly written, and one can implement the proposed approach using the algorithm description in the paper.
- It proposes a new technique to reconstruct 3D shapes from 2D images, which obtains results comparable or better (in terms of standard metrics) than state of the art on a subset of the ShapeNet dataset.
- The proposed reconstruction problem decomposition (described in the previous section) is interesting and novel, albeit not clearly motivated (See paper weaknesses description).

Weaknesses of the paper:
- The paper does not clearly articulate the motivation for the proposed approach, i.e. why it is beneficial to formulate the reconstruction problem as a meta-learning problem. It should also provide a more in-depth comparison of the proposed method with the existing methods which directly predict the implicit shape representation, in order to explain why the proposed formulation outperforms these methods.
- I may be missing it - but nowhere in the paper there is a clear description of how the inference is performed, after the network has been trained. Please add it, e.g., at the end of Section 3.2.
- In the proposed method description at the beginning of Section 3, it is worth clarifying that the output of the Point Generation Network is in fact the training set for the second part of the network, as mentioned in the introduction.
- It would be interesting to perform an ablation study on dimension of the embedding space.
- Figure 10 present an example of shape interpolation. Have the authors tried to interpolate shapes with different topologies and were the results plausible?

Small comments and typos:
- There is a wrong line break at the beginning of Section 5.
- The caption for Figure 11 is too long - it should be shortened and the text should be moved to the corresponding appendix.

**Summary Of The Paper:**

The paper proposes a method for 3D shape reconstruction from image using a bi-level optimization technique. Specifically, the first part of the proposed network extract features from the image and creates a set of 3D points sampled near the surface of the shape to be reconstructed with labels corresponding to the points being inside or outside the 3D shape boundary. Then, the produced sets of points are fed to the second part of the network which is trained to approximate an implicit representation of the shape. At the inference time, a 2D surface of the shape can be extracted from the predicted implicit representation.


**Summary Of The Review:**

The paper proposes a novel method, inspired by meta-learning algorithms, for 3D shape reconstruction from images. The proposed method produces results which are comparable or better than the current state of the art approaches. On the other hand, the paper does not provide  a clear explanation why the proposed approach is suitable for the problem of shape reconstruction and why it is expected to perform better than approaches employing a single network. Adding such an explanation, and a more in-depth analysis of the improvements brought by the bi-level optimization will significantly improve the paper. Currently I recommend "weak accept".

---

> ### Author Response · Authors · 2021-11-20
> **Response to Reviewer DXHG**
>
> We are very grateful for the feedback. We incorporated the revisions (in red).
>
> ### Why meta-learning.
> We were always fascinated by the problem of how to best represent a shape as a point set. There are many ways to represent a surface and we wanted to find out more about how to best represent a surface. The idea was to use a network to learn the surface representation to make the method more interpretable. Over the course of the project this idea got extended with other ideas, such as point embeddings, and various ideas from meta-learning.
> We are not the first to connect meta learning and shape analysis, e.g., [1] and [2] (as we discussed in related works). [1] applies MAML[3]-like algorithms to learn weight initialization, and [2] uses hypernetworks to convert image latents to network weights. In contrast to existing methods, our method assumes a strong prior over the learning process space, which says a (watertight) shape can be represented by a small labeled point set (along with a binary classification algorithm). We believe that training with *prior knowledge* is important in order to increase performance. We argue that this prior contributes to the performance of our method as well as its interpretability as we originally set out to do.
>
> [1] Sitzmann, V., Chan, E. R., Tucker, R., Snavely, N., & Wetzstein, G. (2020). Metasdf: Meta-learning signed distance functions. arXiv preprint arXiv:2006.09662.
>
> [2] Littwin, G., & Wolf, L. (2019). Deep meta functionals for shape representation. In Proceedings of the IEEE/CVF International Conference on Computer Vision (pp. 1824-1833)
>
> [3] Finn, C., Abbeel, P., & Levine, S. (2017, July). Model-agnostic meta-learning for fast adaptation of deep networks. In International Conference on Machine Learning (pp. 1126-1135). PMLR.
>
> ### Clear description of inference.
> We give the detailed description of the inference process in the revised appendix (A.9, Algorithm 2), which starts from an image and ends with a triangular mesh. A comprehensive description of the training process can also be found here (A.9, Algorithm 1).
>
> ### Clarification of PointGen.
> We emphasized the design (in red, Section 3) as the reviewer suggested.
>
> ### Embedding dimension.
> We follow the reviewers suggestion with regards to the embedding dimension. However, this process is a bit different from our main paper. First, increasing the embedding dimension makes the point sets linearly separable (with a high probability). Second, we find that 256D embeddings would cause out of memory errors when training on 32GB v100 cards without changing batch size. Third, we found that to stabilize the training process, we have to add an l2 regularizer to the embeddings (we have not found the true reason but this is an empirical solution).  Thus we choose linear ridge regression without kernels and low dimensional (32, 64 and 128) embeddings. The results can be seen in the revised appendix (A.11, Table 6). We can see that the results improved a bit over the current setup. However, due to the memory issue and the interpretability of the model, we still favor the kernel version of our method. Also see an answer to **reviewer ABfL**.
>
> ### Shape Interpolation.
> We give more examples of inter-category shape interpolation (appendix A.8). One of the examples interpolates shapes of different topology (Fig. 13, bottom row). The table shape is genus-4 while the chair shape is genus-0. A genus-4 shape and a genus-0 shape are *not homeomorphic*. Thus there is *no* continuous transformation between them, but the results are reasonable.
>
> ### Typos.
> We fixed the wrong line break and the issue of long caption in Fig. 11.

---

> > ### Comment · Reviewer_DXHG · 2021-11-24
> > **Reply to Authors**
> >
> > Thank you to the authors for adding the clarifications and making the requested changes - this helps to clarify the points I found unclear earlier. I am happy with the revision and change my recommendation to "accept".

---

### Official Review · Reviewer_ABfL · 2021-11-03

**Correctness:** 3
**Technical Novelty And Significance:** 3
**Empirical Novelty And Significance:** 3
**Recommendation:** 8
**Confidence:** 4

**Main Review:**

This is a good work. There are several strengths.
1) The idea that converting shape reconstruction from image into few-shot learning problem is novel.
2) The authors generalize the differential learner with kernelized algorithms.
3) The results are better then previous works.

There are also some problems needed feedback from the authors.
a) The author use EfficientNetB1 as the Feature Network, and how about utilizing other architecture for the Feature Network? Such as ResNet?
b) The author  generalize the differential learner with kernelized algorithms. There should be experimental result that without kernel skills.
c) More (visual) results on other dataset are needed, such as KITTI and Online Products datasets.

**Summary Of The Paper:**

In this paper, the authors provide a novel idea for shape reconstruction from a single image. They  enable the idea and techniques developed in the few-shot learning literature to the problem of shape reconstruction. They construct the meta-train data by a point generation network, and utilize a differential learner to learn the decision boundary.  They model the reconstruct problem by bi-level optimization.  It is an interesting work. The results are also good.

**Summary Of The Review:**

This is a good paper with novel idea. More results are needed to make it better.

---

> ### Author Response · Authors · 2021-11-20
> **Response to Reviewer ABfL**
>
> We would like to thank the reviewer for providing feedback. We incorporated the revisions (in red).
>
> ### Other architectures for the feature network.
> The Feature network in our model can be easily changed to other architectures. We tested multiple architectures as backbones: densenet121, resnet34 and resnet50. The comparison can be seen in the revised appendix (A.10, Table 5). For our method, the results of densenet and resnet are generally worse than efficientnet. However, even with ResNet we outperform other methods.
>
> ### Differentiable learner without kernels.
> We follow the reviewers suggestion by removing the Gaussian kernel in ridge regression to observe the results. In order to make the inside/outside binary classification easier, we use high dimensional embeddings instead of 3D spatial embeddings. We tested 32D, 64D and 128D. Note that we choose the numbers (32, 64 and 128) because of the GPU memory limit. When using 256D embeddings, we could not train the model on a 32GB v100 GPUs while keeping the current batch size. The results can be seen in the revised appendix (A.11, Table 6). We do not observe significant improvement over kernelized learners. However, according to MetaOptNet[1], extremely high dimensional embeddings (16000) can give a large performance boost compared to low dimensional embeddings (1600). We suspect that higher dimensional embeddings (>=256) might be giving larger performance gain. For now we are unable to verify it without changing current methods. Also see the answer to **reviewer DXHG**.
>
> [1] Lee, K., Maji, S., Ravichandran, A., & Soatto, S. (2019). Meta-learning with differentiable convex optimization. In Proceedings of the IEEE/CVF Conference on Computer Vision and Pattern Recognition (pp. 10657-10665).
>
> ### Visual results on real world data.
> We added more visualizations in the appendix (A.7). We show that the model trained on the dataset ShapeNet can also be used in real world data inference without any retraining and fine-tuning. We believe that the results are good, but there is no ground truth data for a more detailed comparison to previous work.

---

> > ### Comment · Reviewer_ABfL · 2021-11-28
> > **Reply to Authors**
> >
> > Thanks for the response. All the problems  I concerned  are solved. I am happy with the revision and make my recommendation as "accept".

---

### Decision · Program_Chairs · 2022-01-20

**Decision:**

Accept (Poster)

**Comment:**

All reviewers recommended accept after author responses. AC doesn't find any reason to overturn this consensus.